# Nutritional Content, Labelling and Marketing of Breakfast Cereals on the Belgian Market and Their Reformulation in Anticipation of the Implementation of the Nutri-Score Front-Of-Pack Labelling System

**DOI:** 10.3390/nu12040884

**Published:** 2020-03-25

**Authors:** Marie Vermote, Stephanie Bonnewyn, Christophe Matthys, Stefanie Vandevijvere

**Affiliations:** 1Department of Epidemiology and public health, Sciensano, Juliette Wytsmanstraat 14, 1050 Elsene, Belgium; marie.vermote@sciensano.be; 2Test Aankoop/ Test-Achat, Hollandstraat 13, 1060 Sint-Gillis, Belgium; stephanie.bonnewyn@test-achats.be; 3Clinical and Experimental Endocrinology, KU Leuven, 3000 Leuven, Belgium; christophe.matthys@uleuven.be

**Keywords:** nutri-score, breakfast cereals, claims, marketing, nutritional content, reformulation, Belgium

## Abstract

Objective: Breakfast cereals are frequently consumed in Belgium, especially among children. We investigated the nutritional content, labelling and marketing of breakfast cereals and the changes in nutrient content and reformulation in anticipation of the implementation of the Nutri-Score front-of-pack label. Design: Pictures were taken of cereal packages. The WHO-Europe nutrient profiling tool was used to classify cereals into ‘permitted’/‘not-permitted’ to be marketed to children, while the nutritional quality was assessed using Nutri-Score. The proportion of cereals with nutrition and health claims and/or promotional characters on the front-of-pack was compared between permitted and not-permitted cereals and between different Nutriscore categories using Chi-squared tests. The average nutrient contents per 100g were compared between 2017 and 2018 using paired *t*-tests. Setting: Belgium. Participants: All breakfast cereals in the major supermarkets (*n* = 7) in 2017 and 2018. Results: Overall, 330 cereals were included. About 77.2% of cereals were not permitted to be marketed to children but, of those, 22.0% displayed promotional characters. More claims (68.9% of all claims) were found on ‘not-permitted’ compared with ‘permitted’ cereals, particularly nutrition claims. Most claims were displayed on cereals with the allocated Nutri-Score A (42.0%) and C (37.0%). A significant reformulation of cereals was found between 2017 and 2018, with reductions in total sugar (−5%) (*p* < 0.001) and sodium (−20%) (*p* = 0.002) and increases in fibre (+3%) (*p* = 0.012) and proteins (+2%) (*p* = 0.002). Conclusion: Breakfast cereals on the Belgian market are predominantly unhealthy and frequently carry claims and promotional characters. Minimal reformulation occurred over one year. Policy recommendations include restrictions on claims and marketing for not-permitted cereals.

## 1. Introduction

Breakfast is generally considered a key component of a healthy diet [1]. For instance, skipping breakfast has been associated with an increased risk of becoming overweight or obese [2,3,4,5,6] and a higher prevalence of cardiovascular diseases and other non-communicable diseases [3,5,6]. The most recent Belgian food consumption survey 2014/2015 indicated that 78% of Belgians consumed breakfast on five or more days a week, with the lowest frequency of breakfast consumption among adolescents [7]. The consumption of breakfast cereals in Belgium is highest among children (6–9 years) and lowest among the adult population [7]. Among European adolescents aged 12–17 years old, 27% of boys and 24% of girls reported consuming cereals for breakfast [8]. Yet, ready-to-eat breakfast cereals (for example, (boxed) extruded cereals and cornflakes but also certain mueslis) usually contain high levels of added sugar and sodium, and thus may contribute to excessive sugar and sodium intakes, in particular among children and adolescents [9,10,11,12]. In addition, breakfast cereals are among the food products most frequently marketed to children [13,14]. Such marketing has been found to influence children’s food preferences [15], resulting in increased purchasing (requests) [13]. Some studies found a positive link between the frequent marketing of unhealthy foods (i.e., foods high in saturated fats, trans-fatty acids, free sugars, or salt (for example breakfast cereals [16]) and risks for overweight and obesity among children through energy over consumption [16,17]. With the increasing prevalence of overweight and obesity among children and adolescents [18], regulations to support parents and their children to make healthier food choices have increasingly been implemented [19]. The WHO Regional Office for Europedeveloped a nutrient profile model, launched in 2015, in order to support countries in regulating unhealthy food marketing to children [20]. Using this nutrient profile system (NPS), the nutritional contents of products get scored against product-specific thresholds in order to decide whether they are permitted to be marketed to children [20]. In Europe, to date, only Denmark, Ireland, Norway and the United Kingdom have used an NPS to implement marketing restrictions to protect children [20]. Outside of Europe, Chile has restricted marketing across media platforms (including on food packages) to children up to 14 years of age of food products exceeding set thresholds (per 100g) for sodium, total sugar, saturated fat or energy content.

Breakfast cereal packages also frequently include health and nutrition claims [21], affecting consumers’ dietary choices by display of messages or claims implying that a foodstuff has certain health or nutritional properties [22].

According to the European health and nutrition claims regulation (Regulation 1924/2006) [23], all permitted claims are listed, and sanctions are applied when infringements are identified, in order to protect consumers. However, this regulation [23] does not restrict the use of either health or nutrition claims on unhealthy food products. Initially, with the publication of the Regulation 1924/2006, a set of nutrient profiles was planned to be developed for this purpose (article 4), but, more than 10 years later, this has still not been carried out. Various studies from New-Zealand, Canada and Europe highlighted that, irrespective of the existence of nutrition and health claim regulations, a large share of claims were still found on ‘less-healthy’ products [24,25,26,27,28].

In addition to these claims and the marketing found on food packages, government-endorsed front-of-pack nutrition labels (FOPL), such as Nutri-Score, the Health Star Ratings, traffic lights and warning labels, are increasingly implemented to guide consumers towards healthier food choices [29]. While there is a lot of research on the impact of FoPL on consumer purchases, studies on impact of such labels on industry behaviours are scarce [30]. There is some research to suggest that such labels may encourage the food industry to reformulate their products towards healthier versions [31,32,33,34]. Apart from the voluntary Convenant Balanced Diet of 2012–2017, in which the umbrella trade association of the Belgian food industry (FEVIA) and its members made some commitments to reducing nutrients of concern for specific food categories, until recently, no regulations regarding food composition and/or labelling had been introduced in Belgium. Yet, in August 2018, the Nutri-Score was introduced by the Minister of Health in Belgium as a voluntary FOPL. If a Belgian food manufacturer or retailer decides to use the Nutri-Score FOPL, it is obliged to display this FOPL on all their food products within a period of two years. Some of the retailers immediately committed to putting the Nutri-Score on their own brand products after the announcement by the Minister. As of April 1^st^ 2019, the Nutri-Score was officially implemented in Belgium [35], taking into account the European regulation no. 1169/2011 on the provision of food information to consumers [36]. The Nutri-Score rates the nutritional content of packaged foods with five colours/letters, from red (least healthy) to green (most healthy).

The score is calculated based on the energy, saturated fat, total sugar, sodium, fruit, vegetable, nut and legume (FVNL) levels, and the protein and fibre content, of food products [37].

The purpose of this study was to investigate the nutritional content, labelling (i.e., display of claims) and marketing (i.e., display of promotional characters) of breakfast cereals on the Belgian market, ahead of the implementation of the Nutri-Score FOPL. Displays of claims and promotional characters on the packages were compared between the permitted and not-permitted cereals and between different allocated Nutri-Score categories (A-B-C-D-E). In addition, this study investigated potential anticipatory changes in nutrient contents and reformulation efforts by the food industry between 2017 and 2018, ahead of the implementation of the Nutri-Score in Belgium.

## 2. Methods & Materials

### 2.1. Data Collection

Barcode, name of retailer, brand name, product name, food composition information (per 100g contents of energy, protein, total fat, saturated fat, carbohydrates, total sugar, fibre and salt/sodium), and a list of ingredients were collected for all breakfast cereals. The data were collected using photographs taken from the front-of-pack, ingredient list and nutrition information panel of breakfast cereals available in seven different Belgian supermarket chains, selected based on market share (i.e., Delhaize, Carrefour, Colruyt, Lidl, Aldi, Intermarché, Albert Heijn). Data were collected by trained fieldworkers, who were part of the Belgian non-profit consumer organisation ‘Test Aankoop/Test Achat’ for both 2017 and 2018, in the same month of the year ahead of the official implementation of the Nutri-Score in Belgium in April 2019.

### 2.2. Classification of Cereals Using the Nutri-Score

The overall nutritional quality of the breakfast cereals was calculated using the Nutri-Score [35]. The Nutri-Score for each food product is computed, taking into account the nutritional content per 100g and ‘positive points’ are applied for the more unfavourable elements, including energy (kJ, 0–10 points), saturated fat (g, 0–10 points), total sugar (g, 0–10 points), and sodium (g, 0–10 points), and ‘negative points’ for favourable elements, including fruit and vegetable, legumes and nuts (%, 0–5 points), fibre (g, 0–5 points) and proteins (g, 0–5 points). The total of the positive (0–40 points in total) and negative (0–15 points) points is computed, yielding an overall score ranging from −15 for the most healthy foods (0 positive points and 15 negative points) to +40 for less healthy foods (40 positive points and 0 negative points). From this overall score, the five categories of nutritional quality of the Nutri-Score (A-B-C-D-E) are derived.

### 2.3. Classification of Cereals According to the WHO Europe Nutrient Profile Model

The WHO Regional Office for Europe nutrient profile model (WHO-EURO) [20] was used to classify the breakfast cereals in products that are ‘permitted’ and ‘not-permitted’ to be marketed to children. In order to be ‘permitted’ to be marketed to children according to the WHO-EURO model, breakfast cereals must not exceed the thresholds for total fat (<10g/100g), total sugars (<15g/100g) or salt (<1.8g/100g).

### 2.4. Classification of Health and Nutrition Claims and Promotional Characters on the Front-Of-Pack of Cereals Using the INFORMAS-Taxonomy

An internationally standardized taxonomy [38], developed by the International Network for Food and Obesity/NCDs Research, Monitoring and Action Support (INFORMAS), was used to classify the different types of claims and promotional characters directed to children, displayed on the breakfast cereal packages. The categorisation of claims and promotional characters can be found in the free available protocol on the INFORMAS website [39]. Only claims and promotional characters visible on the front-of-pack were included. The classification of claims and promotional characters was conducted by two researchers independently. Any doubt or disagreement during the classification process was discussed until consensus.

Claims were classified into: nutrition claims, health claims and ‘other’ claims, with a further subdivision resulting in a total of eight types of claims, namely: health-related ingredient claims (nutrition claim); nutrient content claims (nutrition claim); nutrient comparative claims (nutrition claim); general health claims (health claim); nutrient and other function claims (health claim); reduction of disease risk claims (health claim); other health-related claims (other claim) and environment-related claims (other claim) (Appendix A) [39]. A claim which ‘states, suggests or implies that a food has particular nutritional properties by virtue of its content of an ingredient’ is classified as a health-related ingredient claim. For example, ‘whole grain’ was considered a health-related ingredient claim. However, a claim that a breakfast cereal contained fruit, such as ‘contains blueberries’, was not classified as such health-related ingredient claim, as the amount of that ingredient was not specified. On the other hand, ‘contains five fruits’ was classified as a health-related ingredient claim, because, as the ingredient is quantified, there is an implication that the food has particular nutritional properties. Additionally, promotional characters on the front-of-pack of the breakfast cereals were also classified using the INFORMAS taxonomy into the following categories: cartoons and company-owned characters, licensed characters, sportspersons, non-sport celebrities, and movie tie-ins.

### 2.5. Statistical Analyses

Descriptive statistics were calculated using SAS 9.4. The nutritional content of breakfast cereals per 100 g on the Belgian market was described using the mean (+ standard error and range) energy, fat, saturated fat, total sugar, protein and fibre content, as well as using the WHO- EURO model and the allocated (calculated) Nutri-Score. The proportion of ‘permitted’ and ‘not-permitted’ breakfast cereals, as well as the proportion of ‘A’,’B’, ‘C’, ‘D’ and ‘E’ breakfast cereals with/without health or nutrition claims and with/without promotional characters on the front-of-pack, was calculated. The total number of claims and claim types were determined, as well as the number of products carrying claims for each claim type. *t*-tests were used to compare the mean content of energy, fat, saturated fat, carbohydrates, total sugar, fibre, proteins and salt between cereals on the market in 2017 and those on the market in 2018. Chi-squared tests were used to analyse differences in the frequency of claims and promotional characters displayed on permitted and not permitted cereals and on cereals with different allocated Nutri-Score.

Matching breakfast cereals for 2017 and 2018 were obtained based on corresponding product code, brand name and product name. In order to verify the possible anticipatory reformulation of these cereals (possibly in anticipation of the implementation of Nutri-Score in Belgium), paired *t*-tests were used to compare the mean values of energy, fat, saturated fat, carbohydrates, total sugar, fibre, proteins and salt between 2017 and 2018. In addition, the proportion of cereals with different allocated Nutri-Score and the proportion of permitted and not permitted cereals were compared between 2017 and 2018 using Chi-squared tests. The statistical significance level was set at α = 0.05.

## 3. Results

### 3.1. Nutritional Quality of Breakfast Cereals on the Belgian Market

Overall, detailed information of a total of 320 and 330 different breakfast cereals was obtained for 2017 and 2018, respectively. Table 1 provides an overview of the mean content of energy, fat, saturated fat, carbohydrates, total sugar, fibre, protein and salt per 100g of cereals permitted or not-permitted to be marketed to children for both 2017 and 2018. Of the 320 included cereals for 2017, 258 products (80.6%) were classified as ‘not-permitted’ to be marketed to children, while 80 (25.0%) cereals were classified with Nutri-Score A, 27 (8.4%) with Nutri-Score B, 139 (43.4%) with Nutri-Score C, 73 (22.8%) with Nutri-Score D and 1 (0.3%) with Nutri-Score E. Of the 330 included cereals for 2018, 255 products (77.2%) were classified as ‘not-permitted’ to be marketed to children, while 98 (29.7%) products were classified with Nutri-Score A, 38 (11.5%) with Nutri-Score B, 134 (40.6%) with Nutri-Score C, 59 (17.9%) with Nutri-Score D and 1 (0.3%) with Nutri-Score E.

When comparing the nutritional quality of all included cereals between 2017 and 2018, differences were significant for total sugar (*p* = 0.04). When including only not-permitted cereals, differences between 2017 and 2018 were significant for carbohydrates (*p* = 0.017), total sugar (0.007), fibre (*p* = 0.033), protein (*p* = 0.007) and salt (*p* = 0.035) (Table 1).

When comparing the nutritional quality of cereals on the market in 2018 between the Nutri-Score system and the WHO-EURO model, 44 out of 98 products with Nutri-Score A, 24 out of 38 with Nutri-Score B, 127 out of 134 with Nutri-Score C, and all products with Nutri-Score D (*n* = 59) and E (*n* = 1) were classified as ‘not-permitted’ to be marketed to children (data not shown).

### 3.2. Claims and Promotional Characters Displayed on Breakfast Cereals

On the front-of-pack of 75 of the 330 cereals (22.7%) on the Belgian market in 2018, no claims were found, while on the front-of-pack of 255 cereals at least one claim was found. Across the 255 cereals with claims, a total of 612 claims were found. Of these claims, 437 (71.4%) were nutrition claims, 18 (2.9%) were health claims, and the remaining 157 (25.7%) claims were classified as other claims. Nutrient content claims (*n* = 287; 46.9%) were the most common type of claims found on cereals. The most common health claims (22.7% of claims) were related to cholesterol reduction, while the most common health-related ingredient claims (51.1% of claims) were related to wholegrain, the most common nutrient content claims (39.6% of claims) were related to fibre, and the most common nutrient comparative claims (42.9%) were related to sugar reduction (data not shown).

The ‘not-permitted’ breakfast cereals (*n* = 255 products) displayed more than twice as many claims (*n* = 422 claims) on the front-of-pack compared to the ‘permitted’ cereals (*n* = 190 claims). About 73.0% of claims on the ‘not-permitted’ cereals were nutrition claims, 2.1% health claims and 24.8% were other claims. For ‘permitted’ products, these percentages were 67.9%, 4.7% and 27.4%, respectively (Figure 1).

The frequency of products with nutrition claims was significantly different between the ‘permitted’ (10%) and ‘not-permitted’ (24.5%) breakfast cereals (*p* = 0.009) and the different Nutri-Score categories (*p* < 0.001). For the frequency of products with health claims, only significant differences were found between the ‘permitted’ (33.3%) and ‘not-permitted’ (100%) breakfast cereals (*p* = 0.005), and not between the different Nutri-Score categories (*p* = 0.071).

Figure 2 provides an overview of the proportion of permitted and not permitted breakfast cereals (according to the WHO-EURO nutrient profile model) with and without promotional characters (=FOPL marketing) on the front-of-pack. In total, 62 cereals (18.8%) carried promotional characters on the front-of-pack. Of the cereals not permitted to be marketed to children (*n* = 255), 56 displayed promotional characters on the front-of-pack, while of the cereals permitted to be marketed to children, only six out of 75 displayed such characters (*p* = 0.007) (Figure 2). Breakfast cereals displayed a total of 84 characters, of which 14 were licenced characters and 70 company-owned characters (data not shown). A higher proportion of not-permitted cereals with nutrition claims (24.5%) displayed promotional characters compared to not-permitted cereals without nutrition claims (17.4%) on the front-of-pack (Figure 2).

Breakfast cereals with Nutri-Scores A and C displayed the most claims overall. The majority of the claims displayed on cereals with Nutri-Score A were nutrient content claims (*n* = 119; 46.1%), environment-related claims (*n* = 54; 20.9%) and health-related ingredient claims (*n* = 41; 15.9%). For cereals with Nutri-Score C, the most common type of claims were nutrient content claims (*n* = 98; 43.3%), health-related ingredient claims (*n* = 53; 23.5%) and environment-related claims (*n* = 41; 18.1%) (Table 2). The proportion of cereals with and without nutrition and health claims within the different allocated Nutri-Score categories can be found in Figure 3. About 75% of products without any type of claim displayed on the front-of-pack were cereals with Nutri-Score C and/or D.

### 3.3. Potential Reformulation of Breakfast Cereals in Anticipation of the Nutri-Score Implementation in Belgium

Based on the unique product code and the product name, 275 products were both available in 2017 and in 2018. Table 3 gives an overview of the nutritional content of these corresponding breakfast cereals in 2017 and 2018, and the differences in nutritional content found between both years. The breakfast cereals sold in 2018 were, on average, significantly lower in total sugar and salt content, and significantly higher in fibre and protein content than the same breakfast cereals sold in 2017 (Table 3). Except for total sugar and salt, no reductions larger than 5% and 20%, respectively, were found between the studied corresponding breakfast cereals of 2017 and 2018.

Regarding the Nutri-Score, 34.5% of the 2017 version products obtained a Nutri-Score of A or B, and 22.2% received Nutri-Score D or E (Figure 4), whereas, for the 2018 version of the products, 37.1% and 19.6% obtained a Nutri-Score of A or B and D or E, respectively. These differences between 2017 and 2018 were statistically significant (*p* < 0.0001).

Based on the WHO-EURO model, only 51 products or 18.5% of the 2017 version breakfast cereals were allowed to use marketing strategies for kids, compared to 60 products or 21.8% of the 2018 version of breakfast cereals. Results showed a significant difference in WHO-EURO classification between 2017 and 2018 (*p* < 0.001) (data not shown).

## 4. Discussion

This study investigated the nutritional content, labelling (i.e., display of claims) and marketing of breakfast cereals on the Belgian market and their reformulation in anticipation of the implementation of the Nutri-Score FOPL. More than three-quarters of the included breakfast cereals were ‘not-permitted’ to be marketed to children, but 22.0% of those products carried cartoons or licenced characters targeted towards children on the pack. This is in line with the results of a New Zealand study by Devi et al. [24], where the majority (72%) of promotional characters were featured on ‘less healthy’ breakfast cereals. Similar figures were found in a global study by Kelly et al. [14], which showed, on average, four times more advertisements for not-permitted foods/beverages than for permitted foods/beverages on television. Such marketing has been found to stimulate children to increase their food consumption [19], and, therefore, several studies have concluded that policymakers need to restrict children’s exposure to unhealthy food marketing, as this marketing to children could play a role in the global obesity epidemic [14,15,17,19,40]. To date, only few countries in Europe have used an NPS in connection with marketing restrictions [20]. In Belgium, such restrictions do not exist, hence 19% of the included cereals carried package marketing to children.

In addition, in this study, more claims, in particular nutrition claims, were found on ‘not-permitted’ compared to permitted cereals. This is in contrast with the New Zealand study by Devi et al. [24], where more nutrition claims were found on healthier breakfast cereals. Other studies also found a greater share of nutrition claims on healthier food products [25,26]. However, comparison with these studies is difficult as they used a nutrient profiling system different from the WHO-EURO model. All the aforementioned studies used the Food Standards Australia New Zealand Nutrient Profiling Scoring Criterion (FSANZ-NPSC) [24,25,26], whereas the current study used the WHO-EURO model [20].

As products using the latter model must be cross-checked against different thresholds (i.e., total fat<10g, total sugars<15g or salt<1.6g per 100 g for breakfast cereals) [20] and the FSANZ-NPSC uses a score based on the nutritional values per 100g of energy, saturated fat, total sugar, sodium, protein, fibre and the fruit, vegetables, nuts, and legumes content [26], differences in categorisation were to be expected. This was confirmed by the study of Labonté et al. [41], in which four different nutrient profile scoring systems (NPS) were compared, and revealed that the FSANZ-NPSC was less strict compared to the WHO-EURO model and two other NPSs, when evaluating whether foods are eligible for marketing to children. Therefore, the results of the current study could have been different if another classification NPS, for example the FSANZ-NPSC, had been used. However, when Nutri-Score was used to assess the nutritional quality of cereals, the results were more in line with the above-mentioned studies [24,26]. When considering products with better nutritional quality according to the Nutri-Score, claims were displayed to a greater extent on healthier (i.e., Nutri-Score A or B) compared to ‘less-healthy’ products (i.e., Nutri-Score D or E). The similarities in the algorithms of both the FSANZ-NPSC and Nutri-Score could be at the basis of this better agreement. However, both NPSs were developed with other purposes in mind [37,42], and have not yet been compared with each other.

The large share of nutrient content claims and health-related ingredient claims found on cereals are in line with the previously mentioned Canadian and New-Zealand studies [24,26] and another European study involving five European countries [28], which also found these the most common types of claims [24,26,28]. Another five-country European study found that 64% of included food and non-alcoholic beverage products carried nutrition claims [27]. However, health claims were less common and represented only 3.4% of all claims in this study.

Changes in the composition of the included breakfast cereals (in anticipation of the introduction of the Nutri-Score NPS) were found, as the Nutri-Score of the 2018 breakfast cereals, was significantly different with the Nutri-Score of the 2017 corresponding breakfast cereals. The same was found regarding the WHO-EURO model. Significant differences were found for total sugar, salt, fibre and protein content between 2018 and 2017; with no reductions larger than 5% except for total sugar and salt. It is, however, difficult to attribute these changes to the introduction of the Nutri-Score, as other commitments by manufacturers were ongoing during that time in Belgium that could have led to a product reformulation.

In the period 2012-2016/2017, for example, the Federation of the Belgian Food Industry (FEVIA) and its members aimed at a reduction of −4% total sugar, an increase of +5% fibre and +8.5% whole grains in breakfast cereals in the Convenant balanced diet [43]. The results of the current study could possibly be a result of the commitment taken with this covenant. Earlier research from other countries has shown that the introduction of FoP labels can have a positive influence on reformulation. For example, Kanter et al. [44], conducted a study on 17 food categories between 2015 and 2016 before the introduction of the implementation of the Chilean Law of Food Labeling and Advertising in Chili in 2016. In this Chilean study, minimal reformulation by industry in anticipation of the implementation of the Law was found, with reductions in nutrient content no larger than 5% [44]. In the study of Ni Mhurchu et al. [32], conducted in New Zealand two years after the implementation of the voluntary Health Star Rating System, significant changes were observed in mean energy density, sodium and fibre contents after the introduction of the Health Star Rating System in 2016 between Health Star Rating-labelled products compared with their composition prior to the adoption of a Health Star Rating [32]. Again, the differences were rather small and, additionally, greater on products with the Health Star Rating compared to those without [32]. Further research is needed to thoroughly evaluate the impact of FOP labelling schemes on food reformulation and other industry practices, such as food marketing and prices. In addition, the impact of Nutri-Score on consumer purchases should be evaluated in Belgium. In France, several experimental studies have already been conducted. An intervention in a web-based supermarket simulated shopping situations with FOP nutrition labels affixed on food products. Around 12,000 participants were randomly assigned to one of five exposure conditions: Guidelines Daily Amount, Multiple Traffic Lights (MTL), Nutri-Score, Green Tick, or control (no front-of-pack exposure). The Nutri-Score significantly led to the highest overall nutritional quality of the shopping basket, followed by MTL and Green Tick, compared with the control, for all socio-economic groups [45]. In an experimental supermarket, about 900 participants were recruited and distributed across three conditions: (1) control situation; (2) application of the Nutri-Score on all breakfast cereals, sweet biscuits and appetizers; and (3) introduction of the Nutri-Score accompanied by consumer information on use and understanding of the label. Significantly higher mean nutritional quality was found for sweet biscuits purchased in the intervention combining the label + education, but not for the other food categories [46].

### Strengths and Limitations

A strength of this study is that this is, to our knowledge, the first study that has investigated the nutritional quality of foods with and without different types of claims in Belgium, as well as the first study to evaluate the reformulation of foods, and in particular cereals, on the Belgian market. The current study was, however, limited to breakfast cereals.

Analysing other food categories could have given us a broader insight into the nutritional composition of products and the use of claims and promotional techniques on the Belgian market. Secondly, data collection only started in 2017. As no data before this date about breakfast cereals or any other food product are available, it was not possible to verify the importance of voluntary industry commitments to reduce the total sugar and sodium content of foods during the period of the Convenant Balanced Diet (2012-2017).

## 5. Conclusions

Breakfast cereals on the Belgian market are predominantly unhealthy and frequently carry claims and promotional characters. Minimal reformulation of cereals was found over one year, but it is likely that the actual implementation of Nutri-Score will stimulate further reformulation. Policy recommendations include restrictions on claims and marketing for unhealthy cereals.

## Figures and Tables

**Figure 1 nutrients-12-00884-f001:**
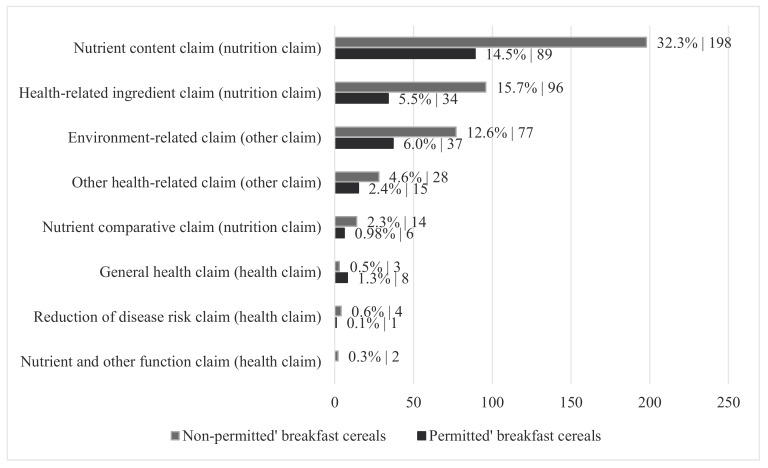
Different types of claims displayed on breakfast cereals on the Belgian market (2018) permitted or not permitted to be marketed by the WHO Europe nutrient profile classification

**Figure 2 nutrients-12-00884-f002:**
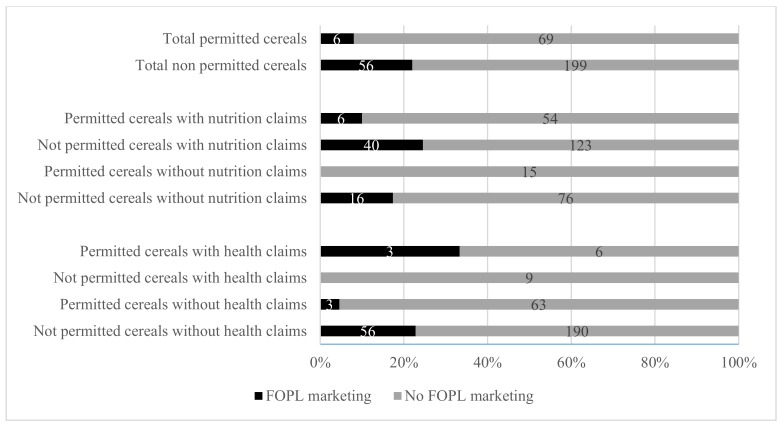
Proportion of breakfast cereals on the Belgian market in 2018 permitted or not permitted to be marketed by the WHO Europe nutrient profile classification with and without promotional characters (=FOPL marketing) on the front-of-pack.

**Figure 3 nutrients-12-00884-f003:**
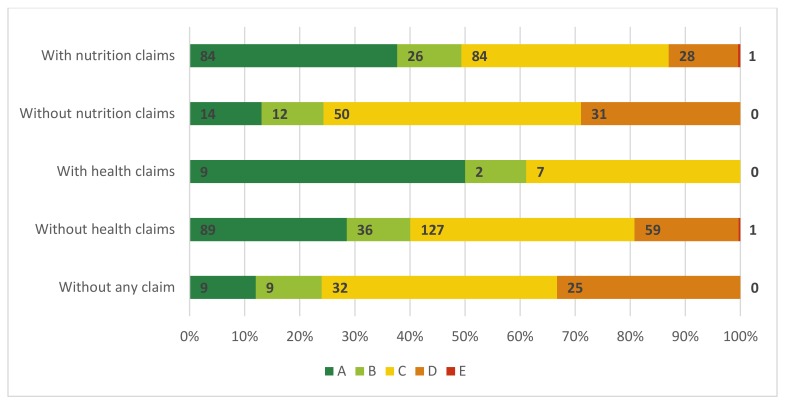
Proportion of breakfast cereals on the Belgian market in 2018 with and without nutrition and/or health claims on the front-of-pack for different allocated Nutri-Score categories.

**Figure 4 nutrients-12-00884-f004:**
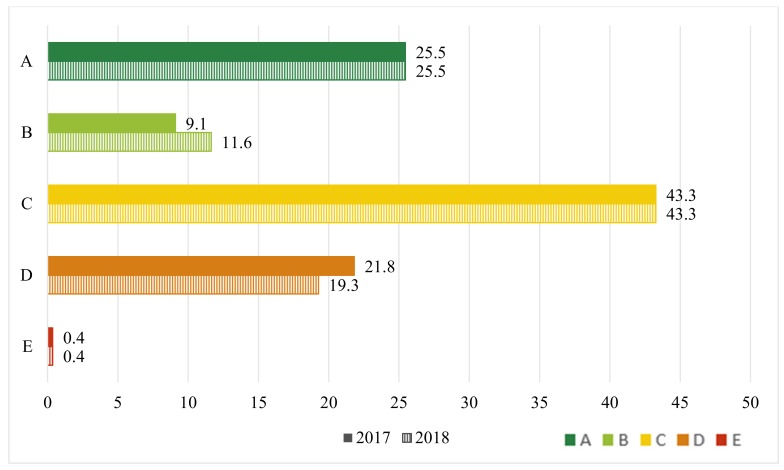
Nutri-Score allocation for breakfast cereals on the Belgian market in 2017 and 2018

**Table 1 nutrients-12-00884-t001:** Mean nutritional content per 100g of the included breakfast cereals (Mean + SE + range) (2017 and 2018)**.**

	2017										2018								
	Breakfast Cereals (*n* = 320)			Permitted Breakfast Cereals (*n* = 62)			Not-Permitted Breakfast Cereals (*n* = 258)			Breakfast Cereals (*n* = 330)				Permitted Breakfast Cereals (*n* = 75)			Not-Permitted Breakfast Cereals (*n* = 255)		
	Mean	SE	Range	Mean	SE	Range	Mean	SE	Range	Mean		SE	Range	Mean	SE	Range	Mean	SE	Range
Energy (kcal)	401.6	2.0	322.0–520.0	371.9	1.8	345.0–423.0	408.8	2.2	322.0–520.0	404.7		2.0	334.0–520.0	374.7	2.0	345.0–420.5	413.5	2.2	334.0–520.0
Fat (g)	8.5	0.3	0.3–25.0	5.4	0.4	0.3–10.0	9.3	0.4	0.3–25.0	9.1		0.4	0.3–32.1	4.5	0.3	0.3–10.0	10.4	0.4	0.3–32.1
Saturated fat (g)	2.5	0.1	0.1–13.9	1.1	0.1	0.1–4.2	2.8	0.1	0.1–13.9	2.4		0.1	0.1–12.0	0.9	0.1	0.1–4.3	2.9	0.1	0.1–12.0
Carbohydrates(g)	68.8	0.5	38.5–89.0	65.6	1.2	53.2–86	69.5	0.6	38.5–89.0	67.5		0.6	34.3–89.0	67.8	1.2	54.2–86.0	67.4*	0.7	34.3–89.0
Total sugar (g)	20.1	0.5	0.7–45.0	7.1	0.7	0.7–15.0	23.2	0.4	1.0–45.0	18.3*		0.5	0.3–54.0	7.1	0.6	0.3–15.0	21.6*	0.5	1.0–54.0
Fibre (g)	7.1	0.2	1.0–27.0	8.4	0.4	1.0–15.0	6.7	0.2	1.5–27.0	7.6		0.2	1.0–27.0	8.3	0.5	1.0–20.6	7.4*	0.2	1.5–27.0
Protein (g)	9.0	0.1	3.4–15.5	11.1	0.3	6.4–15.5	8.5	0.1	3.4–14.5	9.4		0.1	4.5–23.0	10.6	0.2	6.4–14.8	9.0*	0.1	4.5–23.0
Salt (g)	0.5	0.0	0.0–2.9	0.4	0.1	0.0–1.6	0.5	0.0	0.0–2.9	0.4		0.0	0.0–1.9	0.3	0.0	0.0–1.6	0.4*	0.0	0.0–1.9

SE: Standard Error; Range: minimum–maximum. * Significantly different to corresponding mean for ‘2017′.

**Table 2 nutrients-12-00884-t002:** Detailed overview of the different types of claims within each Nutri-Score category (A,B,C,D,E) (2018)**.**

	A	B	C	D	E
Nutrient content claim (Nutrition claim)	119 *19.4%*	34 *5.5%*	98 *16.0%*	35 *5.7%*	1 *0.2%*
Nutrient comparative claim (Nutrition claim)	13 *2.1%*	0 *0.0%*	7 *1.1%*	0 *0.0%*	0 *0.0%*
Health-related ingredient claim (Nutrition claim)	41 *6.7%*	20 *3.3%*	53 *8.7%*	15 *2.4%*	1 *0.2%*
General health claim (Health claim)	7 *1.1%*	1 *0.2%*	3 *0.5%*	0 *0.0%*	0 *0.0%*
Reduction of disease risk claim (Health claim)	1 *0.2%*	1 *0.2%*	3 *0.5%*	0 *0.0%*	0 *0.0%*
Nutrient and other function claim (Health claim)	1 *0.2%*	0 *0.0%*	1 *0.2%*	0 *0.0%*	0 *0.0%*
Other health-related claim (Other claim)	22 *3.6%*	1 *0.2%*	20 *3.3%*	0 *0.0%*	0 *0.0%*
Environment-related claim (Other claim)	54 *8.2%*	6 *1.0%*	41 *6.7%*	13 *2.1%*	0 *0.0%*

**Table 3 nutrients-12-00884-t003:** Nutritional content per 100g in 2017 and 2018 for corresponding breakfast cereals (Mean + SE).

	Breakfast Cereals (*n* = 275)					
	2017		2018			
	Mean	SE	Mean	SE	%-Difference	*p*-Value
**Energy (kcal)**	403.7	35.9	402.7	35.2	−0.2	0.063
**Fat (g)**	8.8	6.4	8.6	6.3	−2.3	0.082
**Saturated fat (g)**	2.6	2.4	2.5	2.3	−4.0	0.066
**Carbohydrates (g)**	68.8	9.9	68.5	10.1	−0.4	0.074
**Total sugar (g)**	20.1	9.2	19.1	9.4	−5.2	<0.001**
**Fibre (g)**	7.0	3.4	7.2	3.7	+2.8	0.012*
**Protein (g)**	9.0	2.1	9.2	2.1	+2.2	0.002*
**Salt (g)**	0.5	0.4	0.4	0.4	−20.0	0.002*

SE, Standard Error. Significantly different to corresponding mean for ‘2017′; * *p* < 0.05; ** *p* < 0.001.

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
