# Peer review of "Nutritional Content, Labelling and Marketing of Breakfast Cereals on the Belgian Market and Their Reformulation in Anticipation of the Implementation of the Nutri-Score Front-Of-Pack Labelling System"

_nutrients, 2020, doi:10.3390/nu12040884_

Round 1
Reviewer 1 Report
Interesting product survey and evaluation of nutrition composition/claims using various nutrient profiling tools (WHO and nutri score).
Abstract: Line 15 describes one aim to investigate “reformulation”, yet line 26 onwards describes comparison of nutrient content between 2017 and 2018. Suggest reformulation is changed to differences in “nutrient contents” over time. Suggest such differences are given with units. For example, is total sugar content given per serving? Or per 100g (as appears to have been done here once I’ve read your methods)? Not clear if you are taking into account changes in portion sizes as your measure of reformulation. 95% CIs rather than p values for changes is also suggested in line 27 onwards. (Also suggest Cis are reported in line 185 onwards in your Results section).
In line with this suggestion, lines 97-98 also state “potential anticipatory reformulation efforts”…, suggest this is more accurately changes in products’ nutrient content from 2018 2019, prior to the implementation of the underscore, which could reflect reformulation? Line 287 is reflect this more accurate wording, in line with the comments above.
You state breakfast cereals may decrease nutritional quality of children’s diets (line 47) , but your cited ref 12 looks to report the opposite. Perhaps rephrase as contribute to excessive intakes of sugars?
12 Michels N, De Henauw S, Beghin L, et al. Ready-to-eat cereals improve nutrient, milk and fruit intake at breakfast in European adolescents. Eur J Nutr [Internet]. 2016;55(2):771–9.
Methods:
You collected data by hand, from in-store? When (month?). Is it possible that some 2017 products, due to launch dates and use-by dates were still on the shelves in 2018? Did you acknowledge this in your limitations? Did you check online for new launches? Did you match all your products or were any unmatched at the 2nd time point?
So you have no data collected on the products’ serving/portion size (in grams) i.e. as displayed on the label? All per 100g?
Line 141- where is the supplementary information? Are Tables and figures intended to appear in Results?
Line 175 and 153: should these make clear content were per 100g?
Lines 217 onwards- suggest giving some idea of proportions products with claims in your sentence text, rather than just p values. Throughout your Results, p values appear more than more descriptive statistics which would be nicer here.
Line 196- why aren’t these data shown. Also, can you give an idea of the which nutrition claims (i.e. source of calcium, iron etc) where the most common show. Same for health claims?
In Figures, good to state what the “permitted” relates to- i.e. based on the WHO model. I got a bit confused and thought you meant the products did not meet conditions for use of the specific (i.e.) nutrition claims. This aspect (presume you weren’t checking this) probably needs making clear in your methods.
Figure 3- would this be better as stacked bar chat to show differences in numbers of products in each group?
Figure 4 state what the nutriscore colours mean in the legend.
Suggest better wording of the Conclusion to reflect aims of the study and results presented….For example, consumer views/perceptions aren’t shown here so its not possible to state about the impact of claims/characters vs of nutriscore for “consumers” (children? Adults?). Suggest sticking to the aims. Potential for this though and warrants future study? Also state reformulation minimal when suggest changes to nutritional composition of products in the time frame… As you know, reformulation might be continuing, gradually, and include portion size reductions potentially, beyond that date.
Reviewer 2 Report
The manuscript evaluates the nutritional profile, labelling, and marketing of breakfast cereals (N = 330) on the Belgian market in to consecutive years with the aim to establish policy recommendations on claims and marketing for not-permitted cereals. Authors conclude that only a minimal reformulation (reduction of sugar and sodium, and increase of fibre and protein) occurred over two years. The investigation is well designed and structured. In addition, the introduction is balanced and provides the necessary background to the topic. The results are well described and discussed. It is recommended publication after very minor comments:
- When comparing nutritional profile from the 2017 and 2018 dataset, it is not clear if differences in the nutritional target are obtained from the average of the more than 300 samples or sample by sample, brand to brand.
- It would enrich the report to include a discussion concerning the type of cereal used in the formulations, besides the use of wholegrains or refined, or the use of non-traditional grains, or even pseudo-cereals.
- It should be per 100g since only breakfast cereals are under study.
- Table 3. Units for energy density should be kJ instead of kcal according to Nutri-Score.
